# The Genetic and Biological Basis of Pseudoarthrosis in Fractures: Current Understanding and Future Directions

**DOI:** 10.3390/diseases13030075

**Published:** 2025-03-03

**Authors:** Amalia Kotsifaki, Georgia Kalouda, Sousanna Maroulaki, Athanasios Foukas, Athanasios Armakolas

**Affiliations:** 1Physiology Laboratory, Medical School, National and Kapodistrian University of Athens, 11527 Athens, Greece; amkotsifaki@med.uoa.gr (A.K.); gkalouda@med.uoa.gr (G.K.); msousanna@uth.gr (S.M.); 2Third Department of Orthopaedic Surgery, “KAT” General Hospital of Athens, 2, Nikis Street, 14561 Kifissia, Greece; afoukas1@otenet.gr

**Keywords:** pseudoarthrosis, non-union fractures, genetic biomarkers, bone healing, biological biomarkers, mesenchymal stromal cells, MSCs, regenerative medicine

## Abstract

Pseudoarthrosis—the failure of normal fracture healing—remains a significant orthopedic challenge affecting approximately 10–15% of long bone fractures, and is associated with significant pain, prolonged disability, and repeated surgical interventions. Despite extensive research into the pathophysiological mechanisms of bone healing, diagnostic approaches remain reliant on clinical findings and radiographic evaluations, with little innovation in tools to predict or diagnose non-union. The present review evaluates the current understanding of the genetic and biological basis of pseudoarthrosis and highlights future research directions. Recent studies have highlighted the potential of specific molecules and genetic markers to serve as predictors of unsuccessful fracture healing. Alterations in mesenchymal stromal cell (MSC) function, including diminished osteogenic potential and increased cellular senescence, are central to pseudoarthrosis pathogenesis. Molecular analyses reveal suppressed bone morphogenetic protein (BMP) signaling and elevated levels of its inhibitors, such as Noggin and Gremlin, which impair bone regeneration. Genetic studies have uncovered polymorphisms in BMP, matrix metalloproteinase (MMP), and Wnt signaling pathways, suggesting a genetic predisposition to non-union. Additionally, the biological differences between atrophic and hypertrophic pseudoarthrosis, including variations in vascularity and inflammatory responses, emphasize the need for targeted approaches to management. Emerging biomarkers, such as circulating microRNAs (miRNAs), cytokine profiles, blood-derived MSCs, and other markers (B7-1 and PlGF-1), have the potential to contribute to early detection of at-risk patients and personalized therapeutic approaches. Advancing our understanding of the genetic and biological underpinnings of pseudoarthrosis is essential for the development of innovative diagnostic tools and therapeutic strategies.

## 1. Introduction

Fractures of long bones are among the most common orthopedic injuries, with significant implications for public health [1]. Approximately 10–15% of these fractures result in non-union or delayed healing, leading to substantial patient morbidity [2]. Non-union, defined as the failure of a fracture to heal within the expected timeframe, often requires repeated surgical interventions, causing persistent pain and disability [3]. Pseudoarthrosis, a severe manifestation of non-union, involves the formation of a fibrous or cartilaginous gap at the fracture site, mimicking a false joint. This condition arises from a complex interplay of biological, mechanical, and systemic factors disrupting the healing environment [4]. Patients with pseudoarthrosis experience crucial functional impairment, reduced quality of life, and prolonged disability, with clinical symptoms including persistent pain, joint stiffness, and deformity. The socioeconomic impact is equally notable as treatment often requires costly surgical interventions and extended rehabilitation [5].

Despite advances in understanding the pathophysiology of bone repair, diagnostic methods remain reliant on clinical and radiographic assessments. This creates ambiguity in determining the optimal timing and approach to treatment [6]. Emerging research has revealed the potential role of biological and genetic biomarkers in predicting fracture healing outcomes, providing a new avenue for precise diagnosis and treatment. Biomarkers linked to blood flow, vascular density, angiogenesis, and angiogenic factors are particularly critical during fracture closure as these elements govern the cellular and molecular processes essential for bone repair [7].

The pathophysiology of pseudoarthrosis underscores its multifaceted nature. Atrophic pseudoarthrosis—characterized by inadequate vascular supply and diminished osteogenic activity—contrasts sharply with hypertrophic pseudoarthrosis, where excessive but unstructured callus forms due to mechanical instability [8]. This variation attracts attention to the complex biological and mechanical factors that affect fracture healing, including the function of osteogenic pathways and vascularization [9]. The disruption of angiogenesis, which is essential for the formation of new blood vessels at the fracture site, further exacerbates healing failure. Impaired vascular density prevents the delivery of oxygen, nutrients, and regulatory molecules necessary for effective bone regeneration [10,11].

Moreover, the epidemiology of pseudoarthrosis varies across fracture types. Femoral shaft fractures, common in polytrauma cases, have a non-union incidence of 1.9–5%, whereas tibial fractures show higher rates of approximately 13.3%, likely due to the tibia’s limited vascularity and exposure to mechanical stress [12]. Humeral shaft fractures are less common but still susceptible to delayed healing, with non-union rates ranging from 1.9% to 10%. These differences underscore the significance of site-specific biological and mechanical factors in determining fracture healing outcomes [13].

The methodology for this review involved a meticulous analysis of the literature from reputable databases, including PubMed, Scopus, and Google Scholar. Studies published over the past two decades were prioritized to ensure relevance and recency. Key terms, such as “pseudoarthrosis”, “non-union fractures”, “bone healing”, “biomarkers in pseudoarthrosis”, “biological factors in bone healing” and “genetic factors in bone healing”, were used to identify pertinent research. This comprehensive approach allowed for the inclusion of clinical studies, systematic reviews, and investigations into advanced diagnostic and therapeutic modalities.

This review explores the genetic and biological mechanisms underlying pseudoarthrosis, emphasizing the critical role of blood flow, vascular density, angiogenesis, and angiogenic factors in bone healing. Vascularization is vital for effective fracture repair as it regulates the supply of cells, nutrients, and growth factors required for osteogenesis. Angiogenic factors, including vascular endothelial growth factor (VEGF) and angiopoietins, are key mediators of this process, promoting the formation of new blood vessels at the fracture site. Impaired angiogenesis is strongly associated with delayed healing and non-union, making it a focus of this review [14,15].

In addition to vascularization, genetic biomarkers play a pivotal role in determining fracture healing capacity. Polymorphisms in genes regulating bone morphogenetic proteins (BMPs), VEGF, and matrix metalloproteinases (MMPs) have been linked to non-union [16]. For instance, studies demonstrate reduced BMP-7 expression and increased levels of its inhibitors, such as Chordin and Gremlin, in non-union tissues. These findings highlight the genetic predispositions that influence the molecular pathways critical for bone repair [17]. Furthermore, microRNAs (miRNAs), such as hsa-miR-149 and hsa-miR-221, emerge as regulators of fracture healing as they modulate the expression of osteogenic genes. Their role in suppressing essential osteogenic markers, such as ALPL and BMP2, further underscores their importance in non-union pathogenesis [18,19].

Clinical management of pseudoarthrosis presents significant challenges. Traditional diagnostic methods rely on radiographic evidence of non-healing and clinical symptoms, including persistent pain and functional limitation [20]. The integration of biomarkers offers a promising alternative, enabling earlier detection of at-risk fractures. Circulating bone turnover markers (BTMs), such as sclerostin (SOST), Dickkopf-1 (DKK1), and osteocalcin, are being explored for their ability to monitor healing progress and predict outcomes [21]. BTMs reflect the dynamic balance of bone formation and resorption, providing real-time insights into the healing environment. By identifying patients with disrupted healing processes, these markers have the potential to guide personalized therapeutic interventions [22].

Therapeutic strategies for pseudoarthrosis range from conservative approaches such as physiotherapy and braces to advanced surgical and biological interventions. Surgical options include bone grafting external fixation, intramedullary nailing, and plating [12]. Autologous bone grafts, a standard treatment, provide osteogenic cells to enhance healing; however, their main limitations are donor-site morbidity and variable efficacy. Innovative approaches—such as low-intensity ultrasound, electromagnetic fields, and extracorporeal shockwave therapy—are gaining acceptance as adjuncts to traditional treatments, promoting bone regeneration through mechano-transduction and enhanced cellular activity [23]. Advances in biological therapies have revolutionized the treatment landscape for pseudoarthrosis. Mesenchymal stromal cells (MSCs) are particularly promising for their ability to differentiate into osteoblasts and enhance angiogenesis through paracrine signaling [24]. Growth factors, such as VEGF and platelet-derived growth factors (PDGFs), are being utilized to stimulate vascularization and bone regeneration [25]. Additionally, circulating biomarkers related to vascular health, including angiopoietins and hypoxia-inducible factors (HIFs), are emerging as potential tools for monitoring and treating non-union fractures [6,26]. These markers reflect the intricate interplay between angiogenesis and osteogenesis, emphasizing the need for integrated therapeutic strategies [27].

In conclusion, pseudoarthrosis remains a significant challenge in orthopedic practice, necessitating a multidisciplinary approach to diagnosis and treatment. Understanding the role of blood flow, vascular density, angiogenesis, and biological and genetic factors in fracture closure is critical to improving therapeutic outcomes [27]. Advances in molecular biology and genetics have highlighted the potential of biomarkers to predict healing success and guide personalized treatments [22]. By synthesizing current knowledge on the genetic and biological underpinnings of pseudoarthrosis, this review aims to provide a foundation for developing targeted therapies and improving patient outcomes. Future research should focus on large-scale studies to validate emerging biomarkers and explore personalized medicine approaches, paving the way for more effective and efficient management of this debilitating condition.

## 2. Biological and Molecular Factors in Non-Union Fractures

As mentioned above, the pathophysiology of pseudoarthrosis is multifactorial, involving an array of biological factors that disrupt the normal healing process [27]. These factors include disturbances in blood flow, MSC activity, angiogenesis, inflammatory responses, cellular senescence, and growth factor signaling [16]. Additionally, blood-based biomarkers and molecular regulators, including miRNAs, have been recognized as critical contributors to the development of non-union and may offer potential predictive tools for patient management [28].

Vascularization is critical for delivering oxygen, nutrients, and signaling molecules essential for osteogenesis [29]. Angiogenesis, the formation of new blood vessels, is a cornerstone of fracture repair, orchestrated primarily by VEGF and angiopoietins [30]. VEGF is responsible for promoting endothelial cell proliferation and migration, which leads to the formation of new capillaries at the fracture site [11]. Impaired angiogenesis is a hallmark of non-union as it leads to inadequate vascular supply, resulting in delayed healing. Investigations have shown that restoring angiogenesis by increasing VEGF or other angiogenic factors improves fracture healing and reduces the risk of non-union [15]. The importance of angiogenesis underscores its potential as a therapeutic target in managing pseudoarthrosis [31].

Beyond vascularization, MSCs have a vital role in bone repair due to their ability to differentiate into osteoblasts and other bone-forming cells [32]. MSCs are recruited to the fracture site, where they proliferate and differentiate into osteoblasts, contributing to the formation of new bone tissue [33,34]. However, in non-union fractures, MSC function is often impaired due to factors such as cellular senescence. Senescence is a process where cells permanently stop dividing in response to stress, damage, or excessive replication [35]. In pseudoarthrosis, MSCs in the non-union tissue exhibit signs of senescence, leading to reduced regenerative capacity [36]. These senescent cells also secrete pro-inflammatory cytokines and matrix-degrading enzymes, which further inhibit healing. Cellular senescence is, therefore, a key factor contributing to the development of non-union [35]. Exploring strategies to prevent or reverse MSC senescence could potentially improve fracture healing and prevent pseudoarthrosis [34].

Apart from the above, BMPs—such as BMP-2 and BMP-7—are critical signaling molecules that promote osteogenesis [37]. These proteins are essential for the differentiation of MSCs into osteoblasts and the formation of new bone tissue. In non-union fractures, the expression of BMPs is often reduced; meanwhile, the expression of BMP inhibitors, such as Gremlin and Noggin, is elevated [38]. This imbalance in BMP signaling disrupts the normal bone repair process, leading to impaired healing [9,39]. Exogenous applications of BMPs have been shown to enhance bone formation and accelerate healing in non-union fractures. BMP signaling pathways, therefore, offer an important avenue for therapeutic intervention in pseudoarthrosis [40]. Understanding the molecular mechanisms regulating BMP expression and activity is vital for developing strategies to restore normal bone healing [40,41].

Furthermore, transforming growth factor-beta (TGF-β) is another key cytokine involved in fracture healing. TGF-β regulates various cellular processes, including the proliferation and differentiation of MSCs [42]. It is particularly important in the early stages of healing, where it promotes the recruitment of MSCs to the fracture site and enhances their differentiation into osteoblasts [34]. However, excessive TGF-β signaling might lead to fibrosis and impaired osteogenesis, as seen in hypertrophic non-unions. An imbalance in TGF-β activity, where excessive signaling hinders bone formation, may contribute to non-union development [43]. Therefore, modulating TGF-β signaling could be a promising therapeutic strategy for enhancing bone healing and preventing pseudoarthrosis [44].

In addition, cellular activities such as growth, differentiation, migration, apoptosis, and proliferation are all influenced by Wnt pathways [16]. Wnt signaling has played an essential part in osteoblast mineralization and differentiation during fracture repair. Wnt activates two major signaling cascades: the canonical Wnt/β-catenin pathway and the non-canonical Wnt/Ca^2+^ pathway [45]. Bone fracture repair is significantly regulated by the BMP/TGF-β signaling pathway. Both TGF-β and BMPs are growth factors. Both canonical and non-canonical signaling cascades are started by their interaction [41]. Bone formation, osteoprogenitor proliferation, and osteoblast differentiation are all influenced by the non-canonical BMP/TGF-β signaling pathway [46]. Among these are signaling molecules that are members of the MAPK (mitogen-activated protein kinase) family. Cytoplasmic serine/threonine kinases are known as MAPKs [16]. Another treatment strategy for fracture healing is the use of biological molecules, such as BMPs, for bone regeneration [41,47].

Notably, BMP expression was lacking in the extracellular matrix and found to be low in the bone ends and canal contents of the non-union location [16]. In vitro tests of BMP’s effects on non-union cell cultures have also shown that adding BMP improves osteogenic differentiation and raises the ALP levels of osteocalcin expression and mineralization capacity [38]. The crucial subject of maintaining equilibrium between BMP gene expression and its inhibitors (Chordin, Noggin, and Gremlin) has been further clarified by studies [48,49]. Chordin, Gremlin, and Noggin levels were higher while BMP-7 gene expression was lower in both investigations [50].

As described above, biological and molecular factors play crucial roles in fracture healing, with the Wnt signaling pathway emerging as a central regulator of bone formation and regeneration [51]. Key antagonists of this pathway, DKK1 and SOST, exhibit dynamic changes during the healing process [52]. Several studies in humans reveal a close negative correlation between local and circulating levels of DKK1 and SOST, with DKK1 levels significantly reduced and SOST elevated during the early post-trauma phase [21]. Subsequently, DKK1 peaks around two weeks postoperatively, followed by a peak in SOST at eight weeks, suggesting a tightly regulated feedback mechanism [21,52]. Factors such as age and smoking can disrupt this balance, and impaired healing has been associated with persistently elevated SOST levels without a compensatory decrease in DKK1 [53]. Experimental models demonstrate the therapeutic potential of targeting these pathways, with dual inhibition of DKK1 and SOST using antibodies significantly enhancing bone healing, callus strength, and bone density [54]. These findings indicate that DKK1 predominantly influences fracture healing, while SOST primarily regulates systemic bone mass, highlighting the promise of dual inhibition therapies for treating fractures [53].

Complementing these insights, blood-based biomarkers have become invaluable tools for predicting and monitoring fracture healing [7]. Circulating markers such as DKK1, SOST, osteocalcin, and Tartrate-resistant acid phosphatase 5b (TRACP 5b) provide critical information about the balance between bone formation and resorption [6]. Elevated levels of DKK1 and SOST are linked to reduced bone formation, while higher osteocalcin and TRACP 5b levels signify active bone remodeling [55]. Emerging molecules, including B7-1 and placental growth factor-1 (PlGF-1), demonstrate potential in predicting post-surgical complications, including non-union, with elevated levels correlated to increased risk [56]. These advancements underscore the potential of combining molecular insights with blood-based biomarkers to enhance the diagnosis, treatment, and management of fracture healing and associated complications [7].

MMPs are a group of enzymes that play a significant role in ECM remodeling, which is essential for the progression of fracture healing [57]. MMPs degrade the extracellular matrix and facilitate tissue remodeling by removing damaged tissue and creating space for new bone formation. However, excessive MMP activity might impair bone healing and contribute to non-union [58]. In hypertrophic non-unions, MMPs such as MMP-7 and MMP-12 have been shown to bind to and degrade BMP-2, preventing its osteogenic effects [46]. This disruption in BMP signaling further hampers fracture healing. Regulating MMP activity to restore the balance of the extracellular matrix (ECM) remodeling could be a valuable therapeutic approach to treating non-union fractures [59].

Moreover, miRNAs have emerged as critical regulators of gene expression in bone healing. These small, non-coding RNA molecules modulate the expression of genes involved in osteogenesis by targeting messenger RNAs for degradation or inhibiting translation [60]. Several miRNAs have been identified as regulators of fracture healing, with hsa-miR-149 and hsa-miR-221 being among the most studied [61,62]. These miRNAs suppress the expression of osteogenic genes, such as ALPL and BMP2, which are essential for bone formation [63]. The dysregulation of miRNAs in non-union tissue suggests that miRNA-based therapies could be a promising approach to modulate gene expression and promote bone healing [61].

Another cohort study found that both male and female patients who developed complications had elevated blood levels of B7-1 and PlGF-1 [56]. B7-1 demonstrated greater sensitivity in predicting complications, making it a promising biomarker for assessing healing progression [64]. Additionally, the study highlighted the importance of sex-specific risk factors, with alcohol consumption being a significant factor for males and increased BMI and comorbidities being important factors for females [56]. These findings suggest that personalized approaches, considering both biological markers and patient-specific factors, may improve the prediction and management of fracture healing outcomes [56].

Neurofibromatosis type 1 (NF1) has been associated with persistent fibrotic non-unions (pseudoarthrosis) in humans. Spatial transcriptomic analysis of fracture pseudarthrosis tissue from an NF1 patient revealed impaired BMP signaling, which is essential for bone healing [65]. In particular, increased MAPK signaling was observed at the fibrocartilaginous–osseous junction, while BMP pathway activation was absent. These findings suggest that the impaired BMP signaling contributes to the delayed healing seen in NF1-related pseudarthrosis [66]. This insight may inform potential therapeutic approaches, such as the use of BMP2, to address this specific type of non-union in NF1 patients [67].

Shifting focus to more detailed cellular interactions, macrophages, derived from hematopoietic stem cells, play a crucial role in the healing process [68]. In other words, M1 macrophages are involved in the inflammatory phase of fracture healing. Their overactivation can impair healing as observed in rats, while controlled activation, such as low-dose TNF-α administration, has been shown to accelerate healing in murine models [69]. Macrophages secrete key cytokines like interleukin 1 (IL-1), IL-6, IL-12, and TNF-α, which are osteoinductive mediators that support bone repair. M2 macrophages, which are alternatively activated by cytokines, contribute to tissue repair and promote vascularization by producing VEGF [70]. A decline in M2 macrophage function, particularly with age, is linked to reduced vascularization and delayed fracture healing [71].

Additionally, osteoprogenitor cells, originating from the local periosteum and potentially from circulating cells marked by the C-X-C chemokine receptor type 4 (CXCR4), are important in fracture healing [72]. In human cases, osteoprogenitor cells cultured on scaffolds have been used to fill large bone defects, facilitating bony union [73]. However, trauma-related hemorrhage may impair the differentiation potential of these cells, which correlates with the high incidence of non-unions in polytrauma patients. Studies also suggest that osteogenic extracellular vesicles (EVs) from injured brain tissue target osteoprogenitors and accelerate bone healing [74]. Age-related decline in osteoprogenitor cells, particularly in women, has been noted, and lower levels of these cells are found in the iliac crests of patients with non-union compared to controls [68]. NOTCH signaling is essential for osteoprogenitor cells as its disruption leads to abnormal bone healing [75].

Osteoblasts, the cells responsible for bone formation, are derived from MSCs [76]. These cells are particularly important during the middle stages of bone healing. By attaching to RANK/RANKL, osteoblasts may also activate osteoclast progenitor cells [77]. While osteoblasts remain unaffected in the early stages of non-union development, later stages show a significant downregulation of key markers, such as runt-related transcription factor 2 (RUNX2) and osteocalcin (OCN), in murine atrophic non-union models [78]. This downregulation suggests impaired osteoblastic maturation, contributing to delayed healing in non-union fractures (Figure 1) [79]. The role of infections, particularly low-grade infections associated with biofilm formation, is another important consideration in the pathogenesis of non-union [80]. Bacterial biofilms are communities of microorganisms that adhere to surfaces, such as bone or implants, and are protected by a matrix that makes them resistant to conventional antibiotics [81].

In summary, crucial factors contributing to non-union include impaired vascularization, MSC dysfunction, dysregulated growth factor signaling, cellular senescence, excessive MMP activity, and miRNA dysregulation (Table 1, Figure 1) [6,16]. Blood biomarkers, such as B7-1, PlGF-1, and other BTMs, offer promising tools for predicting and monitoring fracture healing [56]. Furthermore, addressing infections and promoting angiogenesis and MSC function may provide effective therapeutic strategies for preventing non-union [27].

## 3. The Vascularization in Non-Union Fracture Formation

Bones depend on the blood flow in both healthy and diseased states. Considering that it promotes bone resorption and callus development—as well as neoangiogenesis and revascularization at the fracture site—it is particularly important during fracture healing [89]. Angiogenesis, the formation of new blood vessels from pre-existing ones, is stimulated after fracture by the local production of numerous angiogenic growth factors, and VEGF is the most critical component of the regeneration of the vascular system at the fracture site [90]. Proper vascularization is a key factor in successful bone regeneration as newly formed microvessels ensure the delivery of essential oxygen, nutrients, and growth factors to the callus tissue [91].

Despite significant research aimed at understanding the mechanisms of non-union formation, failed fracture healing continues to be a frequent complication in orthopedic surgery [15]. Fracture sites with good vascularization and abundant fracture hematoma but an unstable mechanical environment typically result in hypertrophic non-union. In contrast, impaired blood supply combined with localized strain concentration is thought to cause atrophic non-union [92]. These non-unions are traditionally seen as the most refractory to treatment owing to the difficulty of improving angiogenesis and can require multiple surgeries over many years [93]. Successful bone repair requires mechanical stability, osteoprogenitor cells, and proper vascularization. One of the main causes of delayed or unsuccessful fracture healing is inadequate blood flow, and major vascular injuries raise the impaired healing rate to 46%, although the worldwide non-union incidence is only 10% [94]. Furthermore, investigations in animal models show that inhibiting angiogenesis during fracture repair leads to fibrous scar tissue formation, similar to human atrophic non-union [95]. Thus, an improved understanding of vascularization in fractures, union and non-union, is essential for advancing orthopedic care.

### Blood Flow Angiogenesis and Angiogenetic Factors

A dense vascular network delivers oxygen and nutrients to all 206 bones in the human body [96]. Trauma to the musculoskeletal system induces a disruption of the vital vascular network, resulting in acute hypoxia and necrosis of the surrounding bone tissue that leads to an inflammatory response [97]. The blood supply of long bones is provided by three main systems of vessels, including the nutrient system, the metaphyseal–epiphyseal system, and the periosteal system [89]. Reduced blood flow within the bone vasculature results in impaired angiogenesis and osteogenesis as well as a downregulation of notch-signaling of endothelial cells [98]. Certain long bones exhibit a distinct distribution of blood supply, which can be compromised under specific injury-related conditions [99]. Bone-vasculature blood flow can be significantly disrupted by various skeletal and systemic diseases, adversely affecting bone regeneration [89]. These include avascular necrosis of the femoral head, which is characterized by reduced endothelial progenitor cells and blood flow interruption due to endothelial cell membrane damage, leading to ischemic injury and necrotic cell death [100].

Furthermore, postmenopausal osteoporosis is associated with reduced blood vessel volume and lower expression of pro-angiogenic markers. Diabetes Mellitus is linked to microangiopathy, resulting in vasoconstriction and decreased blood vessel supply [15]. Lastly, atherosclerosis causes oxidized lipid formation, which reduces bone mass by increasing anti-osteoblastogenic inflammatory cytokines and decreasing pro-osteoblastogenic Wnt ligands [29]. The number and size of blood vessels determine the local blood flow rate. These two factors are regulated through the processes of angiogenesis and vasomotor function, respectively [96]. Angiogenesis is a key component of bone repair. New blood vessels bring oxygen and nutrients to the highly metabolically active regenerating callus and serve as a route for inflammatory cells and cartilage and bone precursor cells to reach the injury site [101]. The sprouting of endothelial cells is the initial step in angiogenesis. In resting blood vessels, endothelial and mural cells are encased in a basement membrane, preventing endothelial cell migration [102]. When new blood vessels are needed, increased production of angiogenic growth factors triggers endothelial cells to break down the basement membrane through MMPs [103]. These MMPs also release stored pro-angiogenic factors, amplifying the angiogenic response. Several endothelial cells, attracted by these signals, become motile and develop filopodia, acting as “tip cells” that lead to the formation of new vessels, while the following cells are referred to as “stalk cells” [104].

Strong evidence indicates that disruptions in the angiogenic response following skeletal injury can significantly impair bone regeneration. Experimental animal studies have demonstrated that inhibiting vascularization with agents—such as TNP-470, non-steroidal anti-inflammatory drugs (NSAIDs), or fumagillin—hinders fracture repair and may result in atrophic non-union formation [15]. A plenitude of mediators and cytokines that promote angiogenesis include VEGF, hypoxia-inducible factor-1α (HIF-1α), bone morphogenetic protein (BMP), FGF, transforming growth factor-beta TGF-β, platelet-derived growth factor (PDGF-BB), receptor activator of the NF-κB ligand (RANKL), a stimulator of osteoclastogenesis and osteoprotegerin (OPG), and an inhibitor of osteoclastogenesis, which are involved in initiating angiogenesis during bone formation and regulating bone formation [25,105,106].

As mentioned above, VEGF is the key angiogenic growth factor essential for fracture repair. It is secreted by various cells, including endothelial cells, macrophages, fibroblasts, smooth muscle cells, osteoblasts, and hypertrophic chondrocytes (Figure 1) [25]. Proper VEGF signaling is crucial for bone repair, as inhibiting VEGF impairs healing, while its local administration enhances bone regeneration [25]. VEGF supports bone repair by stimulating endothelial cells to form new blood vessels, enabling the migration of bone precursor cells to the fracture site, where they differentiate into osteoblasts [76]. In addition, VEGF stimulates endothelial cells to release osteogenic cytokines, which promote the differentiation of progenitor cells into osteoblasts. Lastly, VEGF regulates osteoblast chemotaxis, proliferation, and differentiation, further supporting bone formation [25].

Thus, VEGF serves both indirect and direct roles in bone regeneration, highlighting its importance in successful fracture healing (Table 1 and Figure 1). Experimental studies have demonstrated that VEGF alone cannot initiate the bone regeneration process and that overexpression of VEGF may even hinder fracture healing [107]. These findings suggest that factors beyond VEGF-dependent angiogenesis are crucial for successful bone repair or that angiogenesis during fracture healing is primarily influenced by other growth factors, such as BMP-7 [108]. However, it remains unclear whether VEGF overexpression contributes to non-union formation or if non-union formation triggers a compensatory increase in VEGF levels [15].

## 4. Genetic Factors in Non-Union Fractures

Among the many possible aspects affecting the healing response, some outcomes could be attributed to genetic variations, although a definite role of genetic factors has not been confirmed yet [109]. Numerous studies have used single-nucleotide polymorphisms (SNPs) as markers to evaluate genetic variations within known genes in relation to delayed fracture healing (Table 2).

A study by Dimitriou et al. examined 15 SNPs in genes of the BMP pathway, specifically *NOGGIN* and *SMAD6*, which encode inhibitory molecules of the pathway, and *BMP-2* and *BMP-7*. Out of the 109 patients with long bone fractures that took part in the study, 62 developed atrophic non-union. The results indicated that two of the SNPs and age were associated with the atrophic non-union outcome [110]. Specifically, the *NOGGIN* (rs1372857) G/G genotype and *SMAD6* (rs2053423) T/T genotype have been linked to a higher risk for the development of non-union after fracture [110]. Several studies on mice have also shown that NOGGIN and SMAD6 are very important in bone homeostasis and pathophysiology (Table 2) [109]. Another research was conducted with 167 patients with humeral, femoral, or tibial fractures, 66 of whom developed atrophic non-union. There were some significant differences in the frequency of certain genetic variations between the healed and the non-union groups [111]. More specifically, the *FAM5C* (rs1342913) G allele was found mostly in patients with complete healing, while the *FGFR1* (rs13317) C allele was more common in the non-union group. There was also a noteworthy association between non-union and GTAA haplotype of *BMP4*, with 3.34 times increased risk for the patients with this haplotype (Table 2) [111].

A case-control study based on the Han Chinese population revealed that *NOS2* (rs2297514)—a gene that encodes an enzyme that synthesizes nitric oxide—was significantly associated with the fracture healing process of tibial diaphysis, with the T allele increasing the risk of non-union fracture by 38% in comparison to the C allele. However, this was not statistically significant in the other fracture subgroups that were studied, leading to the speculation that this SNP could be associated with the healing process of the lower limb bones. Still, further research with larger sample sizes needs to be conducted [112]. Another investigation, which explored 144 SNPs within 30 genes associated with fracture healing, concluded that some variations in the NOS2 and IL1B genes may play a role in delayed healing and should be investigated further. Moreover, it was revealed that certain SNPs of the MMP-13 and BMP-6 genes may be protective against non-union (Table 2) [113].

In addition, there is a statistically significant association between a PDGF-A haplotype and non-union, indicating that polymorphisms within this gene can lead to increased risk of delayed bone healing [109,114]. The same study by Zeckey et al. showed that the polymorphism of MMP-13 (rs2252070) is strongly associated with uneventful healing of the fracture (Table 2) [114]. To conclude, SNPs in the NOS2, BMP4, IL1β, and FGFR1 genes were found to predispose individuals to fracture non-union, while polymorphisms in MMP13, BMP6, and FAM5C may offer protection [115].

Research exploring whether polymorphisms in genes related to the regulation of antimicrobial responses affect possible infections of the fracture site and the healing process showed that some polymorphisms in TLR4 and TGF-β might make people more prone to infections in the fracture and slow down the healing [116]. Also, the CYR61 gene—whose transcripts are involved in angiogenesis and are essential for the initial healing process of a fracture—has been found to have some genotypes that affect mRNA expression and increase the risk of non-union. Specifically, the heterozygous TG genotype is significant in fracture non-union compared to the homozygous ones (Table 2) [117].

A cohort study on the Northern European population revealed that CALY and TACR1 genes are associated with fracture non-union (Table 2) [118,119]. CALY had the strongest association, with a significant variant (rs2298122) being a risk factor even after controlling for smoking (which was examined as a confounder). Meanwhile, TACR1, which is involved in pain signaling, may influence the body’s response to pain and inflammatory processes and play a role in bone healing [118]. Other studies have shown that there is a difference in gene expression patterns of genes like CDO1, PDE4DIP, COMP, FMOD, CLU, FN1, ACTA2, and TSC22D1 among the many that were examined. In patients with non-union, the expression of these genes was higher [120].

From a clinical perspective, confirmation of the role of genetic variations in fracture healing could help identify patients that are at risk earlier with the use of genetic testing, and enable timely interventions targeting bone healing [109]. A genetic profiling study has found that the contribution of any single gene to the fracture prognosis is small. However, combining genetic data into existing clinical models could significantly improve the accuracy of fracture risk prediction and guide toward more targeted approaches [121]. Epigenetic alterations should also be taken into consideration. One such example is systemic inflammation triggered by the NF-κΒ pathway, which leads to epigenetic changes that may negatively affect fracture healing, while modifications of DNA methylation at the Rbpjk promoter can improve fracture healing by restoring progenitor cell activity [122].

**Table 2 diseases-13-00075-t002:** “Genetic influence of non-union fracture: an analysis of key candidate genes and polymorphisms”: This table presents an overview of the most critical genetic factors associated with the risk of non-union fractures. Several studies investigating single-nucleotide polymorphisms (SNPs) and haplotypes across diverse genes in different populations have been summarized. Key findings include associations of CALY, NOS2, CYR61, BMP4, and other genetic variations with increased non-union risk, while certain SNPs in MMP13 and BMP6 appear protective. These insights deepen our understanding of the genetic determinants influencing fracture healing, with potential implications for personalized treatment strategies.

Genes Investigated	Study Group	Results	Polymorphisms of Interest	Authors
ADAM8, CALY, ECHS1, FUOM, MIR202, MIR202HG, PRAP1, TUBGCP2, ZNF511, AMPD3, TRACR1, ACHE, MIR6875, MUC3A, MUC12, MUC17, SERPINE1, SLC12A9, SRRT, TRIM56, TRIP6, USFP1, ACAT1, ATM, C11orf65, EXPH5, KDELC2, NPAT, ASTN2, RBMS3	(cohort study) 1760 Northern Europeans with upper or lower fractures—131 non-union	CALY gene SNP was one of the most strongly associated with non-union risk TACR1 gene may influence pain reception and healing process	CALY (rs2298122)	[118]
NOS2	1229 Han Chinese patients with long bone fractures—346 patients with non-union vs. 883 union group	An NOS2 SNP was associated with increased risk of non-union (only in tibial diaphysis fracture subgroup)	T allele of NOS2 (rs2297514)	[112]
CYR61	250 patients with non-union vs. 250 healthy individuals	CYR61 heterozygous genotype affects mRNA expression and may be a risk factor that increases chances for non-union	Heterozygous TG genotype and G allele	[117]
CSF1, IL1B, IL6, IL11, TNFSF11, TNFRSF11B, IFN1α, TNF, COL2A1, COL1A1, TGFB1, TGFB2, TGFB3, BMP2, BMP4, BMP5, BMP6, BMP7, BMP8A, MSTN, GDF10, MMP9, MMP13, VEGFA, VEGFC, ANGPT1, PTN, NOS2, ADRB2, PTGS2	62 patients with long bone fractures—33 non-union vs. 29 union group	Variations in IL1B and NOS2 genes may contribute to non-union risk SNPs of MMP13 and BMP6 could be protective against non-union	IL1B (rs2853550) NOS2 (rs2297514) and (rs2248814) MMP13 (rs3819089) BMP6 (rs270393)	[113]
FAM5C, BMP4, FGF3, FGF1O, FGFR1	167 patients with long bone fractures—66 non-union vs. 101 union group	BMP4 haplotype and an FGFR1 SNP associated with non-union risk FAM5C SNP associated with uneventful healing	BMP4 GTAA haplotype FGFR1 (rs13317) FAM5C (rs1342913)	[111]
TLR2, TLR4, CCR2, CD14, CRP, IL-6, IL-1ra, TGF-β	108 patients with non-union (34 with viable bacterial strains) vs. 122 union group (20 with viable bacterial strains)	Some polymorphisms in TLR 4 and TGF-β may lead to impaired pathogen recognition, prolonged pathogen existence in the fracture site, and higher risk of delayed healing	TLR 4 gene 1/W (Asp299Gly) TGF-β gene codon 10 mutant T and T/C allele	[116]
IGF-1, BMP-2, BMP-4, BMP-7, IL1b, IL-2, IL-3, IL-8, MMP-9, MMP-13, PDGF-A, TNF-α	50 patients with non-union (21 femoral and 29 tibial) vs. 44 union group	PDGF polymorphisms seem to be a risk factor for non-union MMP-13 is highly associated with uneventful healing	PDGF-A CCG haplotype MMP-13 (rs2252070)	[114]
BMP-2, BMP-7, NOGGIN, SMAD6	109 patients with long bone fractures—62 patients with non-union vs. 47 union group	Two genotypes (of NOGGIN and SMAD6) were found to be associated with non-union risk	G/G genotype of rs1472857 of NOGGIN T/T genotype of rs2053423 of SMAD6	[110]

There are several other factors, which differ from patient to patient, that could possibly affect the outcome of a fracture, such as age, sex, lifestyle choices, medication use, and underlying health conditions [27]. Age may be a risk factor, depending on the type of fracture, as it was found to be predictive for clavicle fracture healing capacity but not for humerus fractures [3]. Interestingly though, research has shown that the non-union rate in fractures is higher in patients around 30–44 years old compared to those above 75 years old [123]. Older age is also associated with other risk factors that compromise the healing progress [3], such as obesity, osteoporosis, rheumatoid arthritis, and diabetes [124]. Regarding sex, there is no clear conclusion about whether it affects the healing ability of fractures and studies show mixed results [3].

Several medications may also hinder bone healing, most commonly corticosteroids, chemotherapy, NSAIDs, and some antibiotics [124]. Excessive alcohol intake and smoking can also impair bone formation. Studies have shown that use of tobacco has a significant effect on bone fusion and is associated with a higher risk of non-union [125,126]. Specifically, smoking has been identified as a risk for non-union in the clavicle, diaphyseal, and tibial fractures [68]. A study performed with rats that were fed a diet mixed with ethanol revealed that the healing of the fractures was delayed and the bones had lower bone density and mineral content [127]. Findings in an inception cohort study in which 309,330 fractures in 18 bones were analyzed, five patient-specific risk factors significantly increased the risk of non-union more than 50% across all bones: multiple concurrent fractures, prescription non-steroidal anti-inflammatory drug and opioid use, open fracture, anticoagulant use, osteoarthritis, and rheumatoid arthritis [128]. Ultimately, successful bone healing seems to be shaped by a combination of biological and external factors.

## 5. Diagnostic and Predictive Biomarkers

The methods currently available for diagnosing non-union fractures after the treatment of traumatic injuries include clinical evaluation (stability and malalignment assessment) and imaging methodologies (X-ray, CT, and PET scans) [129]. These methods are able to provide results only after a considerable amount of time following the original intervention. This delay in corrective measures causes many patients to have mobility issues, persistent pain, and a general change in their lifestyle that might even have financial implications [130,131]. Undoubtedly, it is crucial to find new ways to earlier determine cases where the fracture may progress to non-union or pseudoarthrosis. One such method is the detection of circulating biomarkers, for example, bone turnover markers (BTMs), which are released during the healing process of bones and their levels vary based on the degree and progress of recovery [6].

Some possible markers whose levels have been found to change in response to the healing process include BAP, OC, P1NP, and OPG—which are secreted by osteoblasts—as well as CICP, CTX, and TRACP5b. However, currently, there is not enough data to recommend the use of one of these markers alone for the early detection of non-union [6]. The results regarding BAP as a biomarker are mixed, with some studies suggesting that BAP levels were higher in patients with a better outcome after regenerative treatment [132] and significantly lower in patients with delayed union [133]. Interestingly, Herrmann et al. observed the opposite results [6,134]. As for OC, lower levels were detected in the serum of patients with delayed union [132,133] but another study showed no particular difference in the levels of OC between patients with non-union and healed fractures [6,135]. Levels of P1NP and CTX have also been found to decrease in patients with delayed healing [133,136,137], while CICP and OPG increased in people with good outcomes after treatment [136]. However, a study by Marchelli revealed elevated OPG serum levels in patients with atrophic non-union fractures; still, they did not present with an abnormal state of bone metabolism despite the high serum OPG [135]. Moghaddam et al. also reported that TRACP5b was lower at the second and fourth weeks in patients with delayed healing compared to a normal healing rate [137]. Another cohort study supports the idea that bone formation markers OC and P1NP, as well as growth factors VEGF and PGF-1, are promising regarding their predictive role for non-union [7].

In another study, some different gene transcripts detected in blood serum during the bone healing process were examined. The transcripts included IL8, CCL2/MCP-1, PlGF, TGF-β1, TGF-β2, and VEGF. The results indicated that PlGF levels have potential as a biomarker for the early detection of non-union after fracture [138]. Serum levels of PlGF increased continuously during the healing period in patients with good outcomes but displayed a significant peak and then a rapid fall in patients with non-union. Still, there needs to be further research on this growth factor in order to better understand its prospects as a biomarker for non-union [138]. In a small study (of six non-union and six union cases), where half the patients had type II diabetes, it was found that Annexin A3 (ANXA3) levels were elevated in patients with delayed healing compared to uneventful healing [68,139].

Other research has shown that levels of TGF-β1 and TGF-β2, align with the clinical evidence of bone regeneration and could aid in diagnosing non-union [85]. Specifically, it was revealed that serum levels of TGF-β1 were significantly lower a month after trauma in patients with delayed union. The study, examining 103 patients with long bone fractures, observed an initial increase in TGF-β1 levels up to 2 weeks post-fracture, returning to normal values at about 6 weeks. However, a faster decline was seen in patients with delayed healing, with serum levels of the growth factor being significantly lower in those individuals after 4 weeks [140,141]. This supports the notion that an early drop in TGF-β1 levels is indicative of difficulties in fracture healing [140]. Interestingly, another study showed that different levels of TGF-β1 have different effects on osteogenic differentiation and bone healing. Low levels of TGF-β1 activate smad3, which increases BMP2 expression by binding to its promoter. In contrast, high levels of TGF-β1 inhibit BMP2 transcription and hinder bone formation in mice [142]. Elevated serum TGF-β1 levels, have also been found in patients with non-union fractures and are associated with reduced bone mineral content and slow bone healing [142].

Additionally, a few other growth factors have been investigated, such as FGF-2, FGF-23, and BMP-2. It was found that FGF-2 serum levels were significantly lower in patients with non-union and elevated in patients who had complete bone healing after surgery [130,143]. As for FGF-23, it has been found that its quantity increased threefold after surgery in people who developed aseptic loosening after total hip replacement and, therefore, should be evaluated further for the possibility of being used as an indicative parameter for a non-union risk assessment in the future [144]. A study conducted with 42 patients with long bone fractures, which were treated with shockwave therapy, showed elevated serum levels of BMP-2 as well as TGF-β1, VEGF, and nitric oxide in patients with complete union compared to patients with impaired healing. These results indicate that bone healing promoted by shockwave therapy is possibly associated with higher levels of osteogenic factors [145].

Apart from proteins, miRNAs can play a critical role in regulating many pathologies, including fracture non-union. In a study performed on 80 patients (40 with non-union fracture and 40 as the control group), it was reported that miRNA-133a inhibits the RUNX2/BMP2 signaling pathway. The inhibition of RUNX2 and BMP2—key factors that promote osteogenesis and osteoblast differentiation—by miRNA-133a subsequently negatively affects bone formation and delays fracture healing [146]. Furthermore, Jian et al. identified LIN7A and miRNA-29b-3p—which can be acquired from blood samples—as biomarkers to non-union. LIN7A is a small scaffolding protein that plays an important role in cell polarity, adhesion, and signaling [147]. Its levels are more elevated in healthy bone marrow, while its expression is downregulated in instances where the bone does not heal well. It was revealed that miRNA-29b-3p restricts LIN7A expression and, therefore, negatively affects fracture healing [147]. Other findings indicated that hsa-miR-149, hsa-miR-221, hsa-miR-628- 3p, and hsa-miR-654-5p are elevated and could play an important role in the development of atrophic non-union by repressing various osteogenic target genes [148]. Studies involving rats also support the evidence that miRNAs play a crucial role in fracture healing. Elevated levels of miR-451-5p were found in a rat diabetes model 5 to 14 days after fracture [149]. Another study showed that several miRNA levels were highly elevated in non-union rats, with miR-31a-3p/-5p being the most upregulated [149,150].

However, a few requirements must be fulfilled to guarantee that any of the biomarkers might be applied successfully in clinical practice. A biomarker must be conveniently accessible through a quick and low-cost process, have high sensitivity and specificity, and be able to predict non-union reliably in order to be considered optimal [27]. It is evident that even though there have been quite a few studies regarding possible diagnostic biomarkers, clinical findings and radiographic methods still remain the two most important diagnostic tools for non-union [6]. In the future, clinical studies with larger sample numbers that detail patient and sample characteristics will help to better understand the changes in different molecules during the bone healing process [6].

In addition to biomarker discovery, treatment methods of non-union fractures have also evolved over the years as healthcare providers aim to target the specific causes. Atrophic non-unions are often managed with autologous bone grafts (ABGs), which provide the necessary components for healing. Other alternatives include the use of MSCs, osteoconductive scaffolds, and growth factors like BMPs [3]. BMP-2 and BMP-7 have been approved for clinical use in non-unions to enhance healing as they promote bone formation [151], while the scaffolds are used to support the delivery of MSCs and growth factors, improving the healing process [3]. An interesting study performed with serum samples from 114 patients with long bone fractures examined two important negative regulators of the Wnt pathway, DKK1 and SOST. It was discovered that in younger patients with non-union, SOST was elevated without a corresponding drop in DKK1 levels, leading to the belief that mono-targeted therapies directed to the Wnt pathway may not be efficient [53]. Additionally, research in non-human primates supports the idea that combination therapy targeting DKK1 and SOST may be more beneficial in fracture treatment and prevention [52]. Overall, while traditional treatment methods remain essential, there are many innovative approaches that combine biological and mechanical therapies. The results of non-union fractures may be improved by more individualized therapies brought about by the continuous hunt for prognostic biomarkers.

## 6. Therapeutic Implications

### 6.1. Current Treatment Strategies

With advances in surgical techniques and implants, traditional trauma (e.g., simple fracture) is no longer a clinical disease associated with high disability. However, non-unions (especially those caused by infections and bone defects) have become a significant challenge in orthopedics owing to their high incidence (5–10%) and difficulty in applying effective treatments [92]. The treatment of non-unions is predicated on three distinct considerations: the biology of the non-union site, the stability of the bony elements at the non-union site, and patient-related factors [152]. The “gold standard” treatment that is being used is bone grafting, where either an autograft or an allograft bone tissue is transplanted into the area of the non-union to fill bone gaps, promote healing, and stimulate new bone growth [153]. However, performing bone grafting is not without risks and is associated with donor site morbidity, local hematoma, and remodeling issues of the implanted bone [154].

In some cases, surgeons may perform internal/external fixation using screws, plates, or rods to stabilize the bone and ensure proper alignment [155]. To offer the bone the best chance of healing, this is sometimes paired with bone grafting. These techniques have particular disadvantages in that an external fixation frame would not be possible to be removed for several months, and pin-track infections might need immediate medical or surgical care [156]. In some cases, the non-union may be treated by performing an osteotomy, where the bone is intentionally cut and re-aligned to promote proper healing, or by techniques that involve “reaming” the bone around the fracture site—which, when paired with mechanical stimulation or shock-wave treatment, can eliminate non-healing tissue and promote the growth of new, healthy bone cells [157]. However, by removing bone tissue, these methods might weaken the bone and raise the risk of fracture [158].

Electrical stimulation, such as a pulsed electromagnetic field (PEMF), applies a mild electrical field to the fracture site, which can stimulate bone growth and healing [159]. Bone growth stimulators are devices that can be worn externally and use electromagnetic pulses or ultrasound to promote bone healing [160]. Ultrasound-guided bone stimulation, for example, has been shown to increase bone formation and improve healing rates in non-union fractures; however, this works best for fresh fractures or delayed unions but is not always effective for complete non-unions [161].

### 6.2. Gene Therapy

Transferring genes, or more often cDNA, to patients for therapeutic purposes is known as gene therapy. The direct, local transfer of genes to cells within or surrounding the fracture site is the foundation of gene-therapy techniques [162]. Gene-transfer technologies have the potential to aid healing by permitting the local delivery and sustained expression of osteogenic gene products within osseous lesions [163]. Combinations of many growth factors should have a greater effect on bone healing than a single factor alone since they work at distinct stages of osteogenesis [163]. This has been confirmed in animal models using the gene delivery of BMP-2 and BMP-7, BMP-4 and VEGF, and BMP-4 and TGF-β [164]. An alternative approach to the delivery of cDNAs that encode growth factors involves the delivery of transcription factors associated with osteogenesis, such as RUNX2 and osterix (transcription factor Sp7). Since these proteins are intracellular, they are difficult to transport using conventional protein delivery techniques; nonetheless, gene transfer is a good fit for them [165]. Gene transfer uses viral and nonviral vectors. Nonviral methods (transfection) are simpler and safer but less efficient, while viral methods (transduction) are more effective but complex to produce and carry safety risks. Despite these concerns, viral vectors are widely used in gene therapy due to their superior efficiency [166]. To minimize risks, viruses are modified to remove harmful properties while maintaining their ability to deliver genes to target cells [162].

Additionally, retroviruses and lentiviruses integrate into host DNA for long-term gene expression, which is unnecessary for fractures and may cause mutations [167]. Lentiviruses avoid some issues but raise concerns due to their HIV origins [168]. Adenoviruses and AAVs do not integrate, making them better for temporary gene expression. Adenoviruses are effective but trigger strong immune responses, while AAVs are safer but harder to produce and have limited gene capacity [169]. Nonviral vectors include naked plasmid DNA or DNA combined with carriers like liposomes, polymers, or nanoparticles. Gene transfer efficiency can be enhanced using physical methods like electroporation or sonication [170]. Regardless of the vector, there are two basic strategies for gene transfer: ex vivo and in vivo. For ex vivo delivery, cells are removed from the host, genetically modified outside the body, and then re-implanted. For in vivo delivery, the vector is introduced directly into the host, often by injection or by implantation, in association with a matrix [171].

### 6.3. Stem Cell Therapy and Tissue Engineering

A sufficient supply of MSCs and osteoprogenitors is essential for effective bone regeneration. One common method for delivering osteogenic cells to the repair site involves using bone marrow aspirate from the iliac crest, which also contains growth factors. This minimally invasive technique promotes bone healing and has shown promising results [172]. The main goal is to mimic the natural bone environment in order to promote healing [173]. Cells are crucial for bone tissue engineering, with seed cells including embryonic stem cells (ESCs), induced pluripotent stem cells (iPSCs), MSCs, and osteoblasts. Via paracrine signaling, stem cells secrete growth factors (e.g., BMPs, VEGF, and TGF-β) that promote angiogenesis, reduce inflammation, and stimulate endogenous repair mechanisms [174,175]. MSCs modulate the immune response, creating a favorable environment for bone healing [176]. Osteoblasts have strong osteogenic properties but limited proliferation and availability. ESCs have high differentiation potential but face ethical concerns. MSCs, widely used clinically, are sourced from bone marrow, fat, and other tissues but their extraction is painful and yields low quantities [177]. iPSCs, reprogrammed from somatic cells, avoid ethical issues but pose tumorigenic risks unless differentiated into iPS-MSCs [178]. iPS-MSCs exhibit strong proliferation, differentiation, and safety, with no tumorigenic gene expression; they show promise in treating osteonecrosis and bone defects but have rarely been studied with HA/ZrO_2_ composites [179].

### 6.4. BMPs and Other Growth Factors

Advancements in molecular biology have identified key growth factors that enhance bone healing and regeneration. Among them, BMPs, particularly BMP-2 and BMP-7, are the most studied and have been clinically used since the early 2000s for conditions like non-union fractures, joint fusions, and bone defects. Research is ongoing to develop injectable and targeted delivery methods for BMPs [151]. Other growth factors, including PDGF, TGF-β, IGF-1, VEGF, and FGF, play roles in cell proliferation, chemotaxis, and angiogenesis [180]. These have been tested individually or in combination, sometimes with controversial results. Platelet-rich plasma (PRP) is another approach that concentrates these factors from autologous blood to enhance healing [181]. The use of “orthobiologics”, particularly BMPs, is growing but concerns remain about safety, cost, and ectopic bone formation due to the high growth-factor concentrations. In bone tissue engineering, challenges include optimizing the dosage, sustaining controlled release, and combining multiple factors for better results [163]. Nanoparticle technology shows promise for future precise growth-factor delivery but in vivo bone regeneration is still not fully replicable in the lab [182].

## 7. Discussion

Personalized medicine, which tailors medical treatment to an individual’s genetic profile, is emerging as a transformative approach in bone healing. By leveraging genetic information, clinicians can predict healing potential, optimize treatment strategies, and improve outcomes for patients with fractures, non-unions, or bone defects [121]. With the widespread use of sequencing and genetic association analyses in the study of the genetics of complex diseases, the potential contribution of a number of bone formation and growth-related genes and molecules have been investigated regarding the risk of development of non-union during the fracture healing process, including pro-inflammatory cytokines (IL-1 and IL-6), BMPs, and tumor necrosis factor-alpha (TNF-α) [183]. Genetic testing can identify patients at risk of delayed healing or non-unions due to unfavorable genetic variants, for example, patients with BMP2 or VEGF polymorphisms may benefit from early intervention with growth factor therapies (Table 3) [184].

Moreover, a growing number of SNP-related studies have been conducted to evaluate the different genetic “profile” of fracture patients and, specifically, to assess for genetic variants within known genes involved in fracture healing in patients with or without the development of atrophic non-unions [113]. Genetic profiles can guide the use of drugs like bisphosphonates, teriparatide, or NSAIDs to optimize healing, help to select stem cells with favorable genetic traits for autologous transplantation, and customize scaffolds with growth factors or peptides based on genetic predispositions [110]. Technologies enabling personalized medicine include Genome-Wide Association Studies (GWAS), which can identify genetic variants linked to bone healing [185]. Next-Generation Sequencing (NGS) provides detailed genetic profiles for personalized treatment plans. CRISPR/Cas9 and gene editing provides the potential to correct genetic defects impairing bone repair [186]. Lastly, 3D bioprinting creates patient-specific bone grafts based on genetic and anatomical data (Table 3) [187].

Although bone healing treatments have advanced considerably, problems including sluggish healing, extensive deformities, and non-union fractures still exist. New and developing treatments, such as nanotechnology, biomaterials, stem cell engineering, and enhanced gene therapy offer exciting potential for more effective, faster, and personalized bone regeneration [188]. New approaches to bone healing, such as AI-driven medicine, nanotechnology, and smart biomaterials, promise quicker and more efficient regeneration [189]. Gene therapy, particularly CRISPR-based approaches, could enhance endogenous growth factors, while stem cell therapies—including iPSCs and exosome-based treatments—aim to improve bone repair with patient-specific solutions (Table 3) [186]. Three-dimensional printed scaffolds, self-healing biomaterials, and nanotechnology enable the targeted delivery of growth factors and improved implant integration [190]. AI is transforming customized medicine by streamlining treatment regimens and tracking recovery in real time. Notwithstanding their promise, many treatments have drawbacks, such as safety issues, exorbitant expenses, and legal restrictions (Table 3) [191].

Despite advancements in fracture care, pseudoarthrosis is still a major orthopedic concern that frequently results in long-term disability and functional impairment [12]. The complex nature of pseudoarthrosis has been explored in this study, with particular attention paid to the critical roles that biomarkers, biological and genetic variables, and therapeutic approaches play in the onset and management of the condition [27]. Gaining an understanding of these elements is crucial to improving non-union fracture management techniques, both diagnostically and therapeutically.

Numerous biological mechanisms are involved in the complicated pathophysiology of pseudoarthrosis. Delays in fracture healing are largely caused by inadequate vascularization, mechanical instability, and systemic diseases including osteoporosis and diabetes [9]. Non-union is especially linked to disruptions in angiogenesis, a process that is essential for the development of new blood vessels at the fracture site. For instance, poor healing results have been associated with changed levels of angiogenic mediators and decreased VEGF expression [15]. Several recent investigations demonstrated that VEGF expression correlates with fracture healing, highlighting its potential as a therapeutic target. Numerous examples of other markers, including osteocalcin, are suggestive of bone production and remodeling processes in addition to VEGF, providing information on fracture healing [192]. It has been verified that increased levels of these markers predict non-union, especially when systemic or mechanical causes interfere with their expression [6].

Moreover, a major contributing element to the development of pseudoarthrosis is genetics. Studies have demonstrated that genetic differences in important genes related to osteogenesis and fracture healing, such as BMPs, TGF-β, and MMPs, are strongly associated with non-union fractures [22,68]. Some investigations supported that polymorphisms in the BMP-2 gene were linked to a higher risk of non-union, suggesting that genetic predispositions may influence the bone healing process [119]. Other research has looked at the function of the genes for collagen and ECM, pointing out that changes in these genes can affect the bone’s structural integrity and capacity for self-healing [193]. These results highlight how crucial genetic screening is for determining which individuals may be more susceptible to pseudoarthrosis and adjusting treatment plans accordingly [57].

Epigenetic factors further complicate the genetic landscape of bone healing. Environmental factors such as nutrition, smoking, and physical activity can influence gene expression, potentially modifying the healing response [194]. For instance, research on miRNAs that regulate gene expression has identified specific miRNAs, such as miR-149 and miR-221, that might suppress osteogenic markers and inhibit fracture healing [62]. These epigenetic modifications highlight the dynamic interaction between genetics and the environment, which may offer new avenues for therapeutic interventions [194]. Research on the combination of environmental and genetic variables in bone healing is expanding and may eventually result in customized treatment strategies [195].

The use of biomarkers in pseudoarthrosis holds great potential for both early identification and fracture healing tracking [27]. BTMs, including SOST and DKK1, have shown potential in predicting non-union and assessing the progress of healing. These markers reflect the dynamic balance between bone resorption and formation, providing valuable insights into the environment of fractures [6]. A recent systematic review found that elevated levels of sclerostin were associated with delayed bone healing in patients with tibial fractures, suggesting its utility as a diagnostic tool [68]. Additionally, imaging biomarkers—such as the use of micro-CT and MRI—are increasingly being explored to provide a more comprehensive understanding of fracture healing beyond what is possible with traditional radiographic methods. Combining these imaging techniques with circulating biomarkers could significantly enhance the ability to predict and monitor non-union fractures [196].

The development of regenerative medicine and molecular biology has led to changes in therapeutic approaches for pseudoarthrosis. Many patients continue to benefit from traditional procedures such as intramedullary nailing and autologous bone transplants, although they have drawbacks [197,198]. Bone grafting can be associated with donor-site morbidity, and its success is often variable. To address these challenges, newer biological therapies, including MSC therapy, growth factor delivery, and gene therapy, are showing promise. MSCs, with their ability to differentiate into osteoblasts and secrete angiogenic factors, offer a potential solution for enhancing bone healing [199]. According to a recent study, MSC-based treatments may improve the results of healing by encouraging osteogenesis and angiogenesis in non-union fractures [200]. Likewise, growth factors, such as PDGFs and VEGF, are increasingly being used to promote vascularization and speed up the healing process [201]. Mechanical treatments, including electromagnetic fields and low-intensity ultrasound, are being used in conjunction with these biological therapies to encourage bone repair through mechano-transduction and increased cellular activity [202].

**Table 3 diseases-13-00075-t003:** “Emerging future directions in bone healing”: This table outlines innovative approaches and advanced technologies that are shaping the future of bone healing. These methods aim to improve patient outcomes through personalized treatments, advanced genetic techniques, and integrated therapeutic strategies.

Future Direction	Description	Bibliography
Personalized medicine	Tailoring medical treatment based on an individual’s genetic profile to optimize bone healing outcomes.	[121]
Genetic testing	Identifying patients at risk of delayed healing or non-unions due to genetic variants (e.g., BMP2, VEGF polymorphisms).	[113]
Genome-Wide Association Studies (GWAS)	Identifying genetic variants linked to bone healing and pseudoarthrosis through large-scale analyses.	[185]
Next-Generation Sequencing (NGS)	Providing detailed genetic profiles to guide personalized treatment plans.	[186]
CRISPR/Cas9 and gene editing	Potentially correcting genetic defects that impair bone repair, enhancing endogenous growth factors.	[186]
3D bioprinting	Creating patient-specific bone grafts using genetic and anatomical data for improved implant integration.	[187]
Nanotechnology and smart biomaterials	Delivering growth factors and improving implant integration through advanced, targeted therapies.	[190]
Stem cell therapies (e.g., MSCs, iPSCs)	Utilizing patient-derived cells to promote osteogenesis and angiogenesis in non-union fractures.	[200]
AI-driven medicine	Optimizing treatment plans and real-time monitoring of fracture healing using artificial intelligence.	[189]
Epigenetics and miRNA studies	Exploring environmental impacts (e.g., nutrition, smoking) on gene expression and their effect on healing.	[194]
Combination therapies	Integrating mechanical stimulation (e.g., electromagnetic fields) with biological therapies for enhanced healing.	[195]

## 8. Conclusions

In conclusion, pseudoarthrosis is a challenging condition that requires a multimodal approach to diagnosis and treatment. The use of biomarkers and the combination of biological and genetic elements may improve the management of non-union fractures. Genetic testing and customized treatment plans based on distinct biological profiles may make more potent, targeted medications available for those at risk of pseudoarthrosis. Future research should focus on large-scale investigations to validate novel biomarkers and enhance therapeutic techniques, particularly in the fields of gene-based medicines and stem cell therapy. Research might move toward more tailored, accurate therapies by utilizing advancements in genetics and molecular biology. In the long run, this might reduce the burden of non-union fractures and enhance patient outcomes.

## Figures and Tables

**Figure 1 diseases-13-00075-f001:**
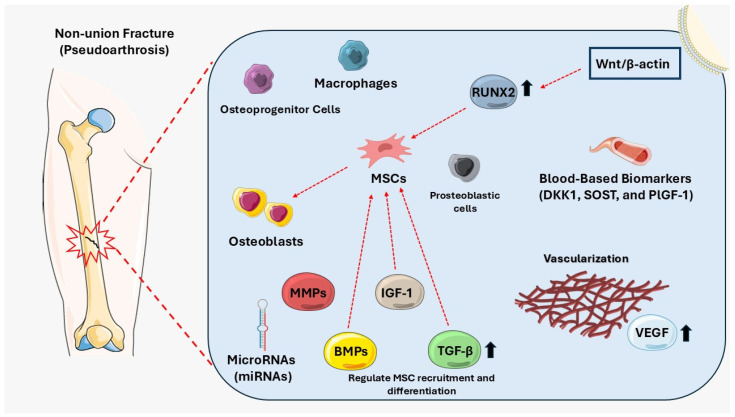
“Key biological factors and Wnt/β-actin molecular pathway in pseudoarthrosis in fractures”: This figure illustrates the most crucial biological and molecular elements involved in pseudoarthrosis, highlighting their roles in non-union fracture. It depicts the interplay between BMPs, MMPs, macrophages, blood factors, and vascularization factors—such as VEGF, TGF-β, and IGF-1—as well as osteoprogenitor cells. Additionally, the figure emphasizes the Wnt pathway’s role in promoting RUNX2 expression in MSCs and pre-osteoblastic cells, ultimately influencing osteoblast function. Understanding these interactions provides insights into potential therapeutic targets for improving bone healing and fracture repair.

**Table 1 diseases-13-00075-t001:** “Key biological factors in pseudoarthrosis and their roles in fracture non-union”: The key biological and molecular elements that contribute to the formation of pseudoarthrosis in fractures are included in this table. Every component is associated with a distinct function in interfering with the typical bone-healing process and causing non-union, offering information about possible treatment targets.

Biological Factor	Role in Non-Union (Pseudoarthrosis)	References
Vascularization	Facilitates oxygen, nutrient, and signaling molecule delivery for osteogenesis. Impaired angiogenesis, often due to VEGF deficiency, results in poor vascular supply and delayed healing.	[25]
Mesenchymal stem cells (MSCs)	Essential for osteoblast differentiation and bone repair. MSC dysfunction or senescence reduces regenerative capacity and creates an inflammatory environment, impairing healing.	[34,82]
Bone morphogenetic proteins (BMPs)	Promote MSC differentiation into osteoblasts. Reduced BMP levels or increased inhibitors (e.g., Noggin, Gremlin) disrupt bone repair and contribute to non-union.	[17,46,50]
Insulin-like growth factor-1 (IGF-1)	Enhances osteoblast proliferation, bone matrix synthesis, and MSC differentiation, promoting bone regeneration. Moreover, IGF-1 upregulates osteocalcin and osteopontin while reducing osteoclast activity, improving fracture healing and reducing non-union risk.	[83,84]
Transforming growth factor-beta (TGF-β)	Regulates MSC recruitment and differentiation. Overactive TGF-β signaling might lead to fibrosis, impairing osteogenesis and contributing to fracture non-union.	[43,46,85]
Blood-based biomarkers	Biomarkers such as DKK1, SOST, and PlGF-1 indicate imbalances in bone formation and resorption. Elevated levels are associated with increased non-union risk.	[21]
Matrix metalloproteinases (MMPs)	Degrade extracellular matrix during healing. Overactive MMPs (e.g., MMP-7, MMP-12) disrupt BMP signaling and impair bone regeneration, leading to pseudoarthrosis.	[58]
MicroRNAs (miRNAs)	Regulate gene expression, essential for osteogenesis. Aberrant miRNA activity (e.g., hsa-miR-149, hsa-miR-221) suppresses bone-forming genes, hindering fracture repair.	[86]
Macrophages	M1 macrophages drive inflammation; M2 macrophages support repair and vascularization. Dysregulated macrophage activity delays healing and fosters non-union.	[87]
Osteoprogenitor cells	Contribute to bone formation. Decreased numbers or impaired differentiation due to aging or trauma correlate with delayed healing and non-union development.	[88]
Osteoblasts	Responsible for bone formation. Impaired osteoblast maturation, marked by decreased RUNX2 and OCN expression, delays fracture healing in pseudoarthrosis.	[76]

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
