# Peer review of "The Genetic and Biological Basis of Pseudoarthrosis in Fractures: Current Understanding and Future Directions"

_diseases, 2025, doi:10.3390/diseases13030075_

Round 1

Reviewer 1 Report

Comments and Suggestions for Authors

In this review, the authors introduced the genetic and biological basis of pseudoarthrosis in fractures. Recent studies about the potential of specific molecules and genetic markers were proposed to serve as predictors of unsuccessful fracture healing. Some novel techniques, mesenchymal stromal cell and Molecular analyses reveal suppressed bone morphogenetic protein were described in detail.

The biological differences between atrophic and hypertrophic pseudoarthrosis were emphasized for targeted approaches to management. This paper provided important and helpful information about the genetic and biological basis of pseudoarthrosis in fractures.

Some contents could be improved to enhance the quality of this paper.

  1. The style of keywords and references must be checked carefully. Some styles of these were incorrect.
  2. In lines 80-82, “This comprehensive approach allowed for the inclusion of clinical studies, meta-analyses, and investigations into advanced diagnostic and therapeutic modalities.”

Meta-analyses apply to quantitative analysis and involve statistical analysis of existing research data. However, this review does not include any quantitative analysis. Please explain.

  1. Remove Figure 1 to the lines 317-318
  2. Please merge section 7, Future Directions, and 8. 8. Discussion, and then please make a table to introduce the future direction to make it more readable.

Author Response

We sincerely thank Reviewer 1 for their thoughtful and constructive feedback on our manuscript, "The Genetic and Biological Basis of Pseudarthrosis in Fractures: Current Understanding and Future Directions." We greatly appreciate their recognition of the value our review provides, particularly regarding the biological differences between atrophic and hypertrophic pseudarthrosis and the exploration of novel techniques and genetic markers. We acknowledge the suggestion to improve certain aspects of the manuscript and have carefully revised the content to enhance its clarity and quality. A detailed point-by-point response to the comments and the revised manuscript will be submitted shortly. Thank you again for the opportunity to strengthen our work, and please extend our gratitude to Reviewer 1 for their insightful comments.

Reviewer 2 Report

Comments and Suggestions for Authors

The manuscript entitled "The Genetic and Biological Basis of Pseudarthrosis in Fractures: Current Understanding and Future Directions" is well written and offer new insights to this relatively common complication, that is long bone pseudarthrosis. The manuscript addresses the questions of what causes pseudarthrosis and what future treatments are under development. It provides a good overview of the multiple factors involved into the process of pseudarthrosis as the authors clearly review all the known biological, vascular and genetic factors involved in pseudarthrosis. The table and the figure are clear and provide a better understanding of the factors involved. It's novelty is that it is the most complete review in the current literature regarding the biological factors involved into pseudarthrosis. A review on genetic and biological factors of pseudarthrosis is an original idea. Another important chapter is the therapeutic implications of the information, including both current treatment strategies, as well as future gene and stem cell therapies. The authors also provide their opinion on future directions  including personalized treatment plans. The references are appropriate. I consider that the manuscript is valuable for both clinicians and researchers in the field. I recommend for publication in current form. 

Author Response

We sincerely thank Reviewer 2 for their thoughtful and positive evaluation of our manuscript, "The Genetic and Biological Basis of Pseudarthrosis in Fractures: Current Understanding and Future Directions." We greatly appreciate their recognition of the novelty and comprehensiveness of our review, as well as their acknowledgment of the value it brings to both clinicians and researchers.

Please extend our gratitude to Reviewer 2 for their kind and encouraging feedback. We are pleased to hear that the manuscript is recommended for publication in its current form.

We will be submitting the revised manuscript and the full responses to the other reviewers shortly.

Thank you for your time and consideration.
